# Third-Order Optical Nonlinearities of 2D Materials at Telecommunications Wavelengths

**DOI:** 10.3390/mi14020307

**Published:** 2023-01-25

**Authors:** Linnan Jia, Jiayang Wu, Yuning Zhang, Yang Qu, Baohua Jia, David J. Moss

**Affiliations:** 1Optical Sciences Center, Swinburne University of Technology, Hawthorn, VIC 3122, Australia; 2School of Science, RMIT University, Melbourne, VIC 3001, Australia; 3Australian Research Council (ARC) Industrial Transformation Training, Centre in Surface Engineering for Advanced Materials (SEAM), RMIT University, Melbourne, VIC 3000, Australia

**Keywords:** third-order optical nonlinearity, 2D materials, telecommunications band

## Abstract

All-optical signal processing based on nonlinear optical devices is promising for ultrafast information processing in optical communication systems. Recent advances in two-dimensional (2D) layered materials with unique structures and distinctive properties have opened up new avenues for nonlinear optics and the fabrication of related devices with high performance. This paper reviews the recent advances in research on third-order optical nonlinearities of 2D materials, focusing on all-optical processing applications in the optical telecommunications band near 1550 nm. First, we provide an overview of the material properties of different 2D materials. Next, we review different methods for characterizing the third-order optical nonlinearities of 2D materials, including the Z-scan technique, third-harmonic generation (THG) measurement, and hybrid device characterization, together with a summary of the measured *n*_2_ values in the telecommunications band. Finally, the current challenges and future perspectives are discussed.

## 1. Introduction

All-optical signal processing based on nonlinear optical devices is an attractive technique for ultrahigh speed signal processing for optical communication systems. It offers broad operation bandwidths, ultra-high processing speeds, together with low power consumption and potentially reduced footprint and cost. Integrated nonlinear optical photonic chips have been based on a few key materials including silicon (Si) [1,2,3], doped silica (SiO_2_) [4,5], silicon nitride (Si_3_N_4_) [6,7], aluminum gallium arsenide (AlGaAs) [8,9,10], and chalcogenide glasses [11,12]. These have enabled a wide range of devices from Raman amplification and lasing [13,14,15], wavelength conversion [5,12,16,17,18], optical logic gates [19,20,21,22], and optical frequency comb generation [23,24,25,26], to optical temporal cloaking [27], quantum entangling [28,29,30], and many others. Despite their success, no platform is perfect—they all have limitations, such as a relatively small Kerr nonlinearity (*n*_2_) (e.g., for Si_3_N_4_) or high two photon absorption (for silicon in the telecommunications band), resulting in a low nonlinear figure of merit (FOM = *n*_2_/(*λβ*_TPA_), with *n*_2_ and *β*_TPA_ denoting the effective Kerr coefficient and TPA coefficient of the waveguides, respectively, and *λ* the light wavelength).

To overcome these limitations, newly emerging materials have attracted significant attention, particularly 2D-layered materials, such as graphene [31,32,33], GO [34,35,36], TMDCs [37,38,39,40], h-BN [41,42,43], and BP [44,45,46], where their atomically thin nature yields unique and superior properties. Compared with bulk materials, the 2D materials possess surfaces that are free of dangling bonds due to their weak out-of-plane van der Waals interactions [32,37]. In particular, their properties are highly dependent on the number of atomic layers—not only is their optical bandgap highly layer thickness dependent but they can also exhibit an indirect-to-direct bandgap transition (and the reverse), which provides powerful ways in which to tune their optical responses [37,46,47,48]. Further, their broadband photoluminescence and ultrahigh carrier mobility are highly attractive features for photonic and optoelectronic applications [33,49,50,51,52,53]. The unique photon-excited exciton and valley-selective properties of monolayer TMDCs and their heterostructures are promising for the development of future spintronic and quantum computing devices [37,38]. Finally, in addition to their linear optical properties, 2D materials exhibit remarkable nonlinear optical properties including strong saturable absorption (SA) [54,55,56,57], a giant Kerr nonlinearity [58,59,60,61,62], and prominent second- (SHG) and third-harmonic generation (THG) [44,63,64,65], opening up new avenues for high-performance nonlinear optical devices.

In contrast to the second-order optical nonlinearity that only exists in non-centrosymmetric materials, the third-order susceptibility is present in all materials, which gives rise to a rich variety of processes, including four-wave mixing (FWM), self-phase modulation (SPM), cross-phase modulation (XPM), THG, two-photon absorption (TPA), SA, stimulated Raman scattering, and many others. These third-order nonlinear optical processes are quasi-instantaneous with ultrafast response times on the order of femtoseconds [66]. This has motivated ultrafast all-optical signal generation and processing for telecommunications, spectroscopy, metrology, sensing, quantum optics, and many other areas [67,68].

In this paper, we review recent progress in the study of the third-order optical nonlinearities of 2D materials specifically in the telecommunications wavelength band near 1550 nm, in contrast with other reviews [44,69,70] that focus predominantly in the visible wavelength range. We discuss the different techniques for characterizing the third-order optical nonlinearity and the prospects for future development. In Section 2, the material properties of different 2D materials are briefly introduced and compared. Next, we review different methods used to characterize the third-order nonlinear optical response of 2D materials, including Z-scan technique, THG measurement, and hybrid device characterization. We also summarize the measured values of *n*_2_ of different 2D materials in the telecommunications band. Finally, the conclusion and future perspectives are discussed in Section 4. Our review is aimed to help readers have a view of the progress in this field and to provide guidance for optimizing the properties and device applications of 2D materials in future optical telecommunications systems.

## 2. 2D Materials

The past decade has witnessed an enormous surge in research on layered 2D materials—many have been discovered and synthesized with a wide range of properties. In this section, we briefly introduce some key 2D materials such as graphene, GO, TMDCs, h-BN, and BP as shown in Figure 1 and discuss their electrical and optical properties.

### 2.1. Graphene and Graphene Oxide

Graphene, and its derivative, graphene oxide (GO), have been intensely studied due to their excellent mechanical, electrical, and optical properties [33,71,72]. Graphene has a gapless band structure, in which the conduction and valence bands meet at the K point of Brillouin zone, resulting in its semimetal nature [31,44,73]. In contrast, GO is an electronically hybrid material, featuring both conducting *π*-states from sp^2^ carbon sites and a large energy gap between the σ-states of its sp^3^-bonded carbons [34,74]. Their unique band structures result in novel electrical and optical properties, where for graphene, for example, the electrons and holes act as massless Dirac fermions resulting in extremely high carrier mobilities (>10^5^ cm^2^/Vs) even under ambient conditions [31]. In contrast, GO exhibits a band gap that is tunable by adjusting the degree of reduction, which in turn affects the electric and optical properties. In addition, GO exhibits fluorescence in the near-infrared (NIR), visible and ultraviolet regions [34,35,36], which is very promising for light emitting devices. Moreover, the excellent nonlinear optical properties of both materials have been reported, including strong saturable absorption (SA) [75,76], a giant optical Kerr nonlinearity [58,59], leading to efficient self-phase modulation [77], FWM [78,79], as well as high harmonic generation [63].

### 2.2. Transition Metal Dichalcogenides

Transition metal dichalcogenides (TMDCs) with the formula of MX_2_ (where M is a transition metal and X is a chalcogen), is another widely studied family of 2D materials. Different to the semimetal graphene, monolayer TMDCs, such as MoS_2_, MoSe_2_, WS_2_, and WSe_2_, are typically semiconductors that have bandgaps from 1 eV to 2.5 eV, covering the spectral range from the near infrared to the visible region [37,38]. Moreover, TMDCs can exhibit a transition from direct- to indirect-bandgaps with increasing film thickness, resulting in strongly thickness-tunable optical and electrical properties. For instance, MoS_2_ exhibits layer-dependent photoluminescence, with monolayer films showing a much stronger photoluminescence [80]. Monolayer hexagonal TMDCs also exhibit unique band structure valley-dependent properties, such as valley coherence and valley-selective circular dichroism [37,81], offering new prospects for novel applications in optical computing and information processing. For the nonlinear optical properties, TMDCs with odd numbers of layers have no inversion symmetry, and so exhibit a non-zero second-order (and higher even-order) nonlinearities that are absent in graphene and even-layer TMDCs [44,64]. Recently, noble metal TMDCs, including PdSe_2_ and PtSe_2_, and PdTe_2_, have also attracted increasing interest in the fabrication of high performance electronic and optical devices, such as ultra-broadband photodetectors [39,40] as well as mode-locked lasers [82].

### 2.3. Black Phosphorus

Black phosphorus (BP) is another attractive single element 2D-layered material which has been widely studied. It has a puckered crystal structure, yielding a strong in-plane anisotropy in its physical properties in the “armchair” and “zigzag” directions, opening new avenues for anisotropic electronic and optoelectronic devices [44,45,83]. Moreover, BP is a semiconductor that features a layer thickness-dependent direct bandgap from 0.3 eV (bulk) to 2.0 eV (monolayer), bridging the gap between the zero-bandgap graphene and large-bandgap TMDCs [48,83]. This broad bandgap tunability is very suitable for the photodetection and photonic applications from the visible to mid-infrared spectra regions [46,47,84]. For the nonlinear optical properties, the layer thickness tunable and polarization-dependent THG and optical Kerr nonlinearity have been demonstrated recently [83,85]. Broadband SA has also been observed in BP, demonstrating its strong potential for ultrafast pulsed lasers [86,87,88].

### 2.4. Other Emerging 2D Materials

A wide range of other novel 2D low-dimensional materials have been investigated, including h-BN, MXenes, perovskites, as well as MOFs, which greatly enriches the family of 2D materials. h-BN is an electrical insulator with a large bandgap of around 5.9 eV [41,89] making h-BN a candidate for ultraviolet light applications. It also has an ultra-flat surface as well as excellent resistance to oxidation and corrosion, which are both highly useful as a dielectric or capping layer to protect the active materials or devices from degradation [41].

MXenes belong to another family of 2D materials, including 2D transition metal carbides, nitrides, and carbonitrides. Typically, the electronic structure of MXenes can be tuned by varying the surface functional groups. For instance, nonterminated Ti_3_C_2_ theoretically resembles a typical semimetal with a finite density of states at the Fermi level, whereas it can transition to a semiconductor when terminated with surface groups, such as OH and F groups [90]. MXnens also exhibit superior optical properties, such as a high optical transmittance of visible light (>97% per nm) [38,91], and excellent nonlinear optical properties [57].

Organometal-halide perovskites have a general formula of ABX_3_, where typically A = CH_3_NH_3_^+^, B = Pb^2+^, and X = I^−^, Br^−^, Cl^−^ or mixtures [92]. Due to their prominent photovoltaic features and luminescence properties, organometal-halide perovskite semiconductors have been widely used to design high performance solar cells as well as light-emitting diodes [51,52,53]. Metal-organic frameworks (MOFs) are organic–inorganic hybrid porous crystalline materials with metal ions or metal-oxo clusters coordinated with organic linkers [93,94]. Thanks to this unique structure, 2D MOFs exhibit enhanced photo-physical behaviour and are promising for various applications, from light emission and sensing to nonlinear optical applications [95,96].

## 3. Third-Order Optical Nonlinearities of 2D Materials in the Telecommunications Band

With their excellent third-order optical nonlinearities, 2D materials are promising functional materials for high-performance nonlinear optical devices. In this section, we review the different methods used to characterize their third-order nonlinear optical response. These include the Z-scan technique, THG measurement, and hybrid device characterization. We also summarize and compare the measured *n*_2_ values of different 2D materials in the telecommunications band.

### 3.1. Third-Order Optical Nonlinearity

The nonlinear optical response of a material in the dipole approximation is given by [1,97]:(1)$$\tilde{P}\left(t\right)=\varepsilon_{0}[\chi^{\left(1\right)}\cdot \tilde{E}\left(t\right)+\chi^{\left(2\right)}\;:\;\tilde{E}\left(t\right)\tilde{E}\left(t\right)+\chi^{\left(3\right)}\vdots \;\tilde{E}\left(t\right)\tilde{E}\left(t\right)\tilde{E}\left(t\right)+\ldots \;]\;\;\;\;\;\;\;\;\;\;\;\;\;\;\;\;$$
where the $\tilde{P}\left(t\right)$ is the material electronic polarization, $\tilde{E}\left(t\right)$ is the incident field, χ^(n)^ are the n^th^-order nonlinear optical susceptibility. The first-order term χ^(1)^ describes the linear refractive index including refraction and absorption and is a result of the dipole response of bound and free electrons to a single photon [1]. The second-order term χ^(2)^ is a third-rank tensor, nonzero only for non-centrosymmetric materials, describes second-harmonic generation (SHG), sum-and difference frequency generation (SFG, DFG), optical rectification, the Pockels effect and others. The third-order nonlinear optical susceptibility χ^(3)^ is particularly important because it exists in all materials regardless of the crystal symmetry and gives rise to a rich variety of nonlinear processes, represented by THG [1,97], FWM [78,98], SPM [61,99], and XPM [100,101]. These form the basis of all-optical processing devices, such as wavelength conversion, optical comb generation, quantum entanglement, and more.

Equation (2) gives a simple description of the relevant third-order nonlinear optical effects corresponding to ${\tilde{P}}^{\left(3\right)}\left(t\right)=\varepsilon_{0}\chi^{\left(3\right)}\cdot {\tilde{E}}^{3}\left(t\right)\;$[97] as follows:(2)$${\tilde{P}}^{\left(3\right)}\left(t\right)=\varepsilon_{0}\int_{-\infty }^{\infty }\frac{d\omega_{1}}{2\pi }\int_{-\infty }^{\infty }\frac{d\omega_{2}}{2\pi }\int_{-\infty }^{\infty }\frac{d\omega_{3}}{2\pi }\chi^{\left(3\right)}\left(\omega_{\sigma };\omega_{1},\omega_{2},\omega_{3}\right)\times E\left(\omega_{1}\right)E\left(\omega_{2}\right)E\left(\omega_{3}\right)e^{-i\omega_{\sigma }t}$$
where $\omega_{\sigma }=\omega_{1}+\omega_{2}+\omega_{3}$, with $\omega_{1}$, $\omega_{2}$, and $\omega_{3}\;$denoting the angular frequencies. Different *χ*^(3)^ effects can be described with different wave frequency combinations, such as THG ($\chi^{\left(3\right)}\left(\omega_{\sigma }=3\omega_{1};\omega_{1},\omega_{1},\omega_{1}\right)$), non-degenerate FWM ($\chi^{\left(3\right)}\left(\omega_{\sigma }=\omega_{1}+\omega_{2}-\omega_{3};\omega_{1},\omega_{2},-\omega_{3}\right)$) and degenerate FWM ($\chi^{\left(3\right)}\left(\omega_{\sigma }=2\omega_{1}-\omega_{2};\omega_{1},\omega_{1},-\omega_{2}\right)$).

A key component of χ^(3)^ is given by n2=3·Re[χ(3)4cn02ε0], which reflects the intensity-dependent refractive index change, known as the Kerr effect and the complex refractive index *n* can be expressed as [1,97]:(3)$$n=n_{0}+n_{2}I-i\frac{\lambda }{4\pi }\left(\alpha_{0}+\alpha_{2}I\right)$$
where *n*_2_ represents the Kerr coefficient or Kerr nonlinearity, *I* is the light intensity, *λ* is the wavelength, *α*_2_ is the nonlinear absorption induced by the third-order susceptibility *χ*^(3)^, and *n*_0_, *α*_0_ are the linear refractive index and absorption, respectively.

In this paper, we focus on *χ*^(3)^ of 2D materials for key nonlinear processes that form the basis for ultra-high speed all-optical signal generation and processing, with response times on the order of femtoseconds [102,103]. These include SPM and XPM, governed largely by *n*_2_ via the Re (*χ*^(3)^), as well as FWM and THG that are mainly governed by the magnitude of |*χ*^(3)^|, although the latter are also sensitive to the complex value of *χ*^(3)^ via phase-matching effects. The *n*_2_ component of *χ*^(3)^ accounts for two-photon absorption (TPA) via the Im (*χ*^(3)^) and can also result in saturable absorption (SA). Both are intrinsic functions of the material’s bandgap, but can also be influenced by free carrier effects. At photon energies well below the bandgap, all *χ*^(3)^ components will become degenerate, but near, or above, the bandgap, they will in general be quite different. Finally, since nonlinear absorption is always present, it will affect the efficiency of all third-order nonlinear optical processes, not just *n_2_*, even though it does not arise directly from other *χ*^(3)^ components such as THG and FWM, for example. Further, these processes will generally scale differently with pump power to n_2_, and so the conventional nonlinear FOM may not be a useful benchmark.

### 3.2. Characterization Methods

#### 3.2.1. Z-Scan Technique

Measuring the Kerr coefficient of a material is needed in order to design and fabricate nonlinear optical devices. The Z-scan method, introduced in the 1990s [104] is an elegant method to measure the third-order optical Kerr nonlinearity of a material. This technique involves open-aperture (OA) and closed-aperture (CA) measurements, which can be used to measure the third-order nonlinear absorption and nonlinear refraction, respectively. CA Z-scan method is widely used to measure the nonlinear refractive index (Kerr coefficient) of an optical material. The valley–peak and peak–valley transmission curves are the typical results of the CA measurement, as shown in Figure 2a. When the nonlinear material has a positive nonlinear refractive index (*n*_2_ > 0), self-focusing will occur which results in the valley–peak transmission curve. The peak–valley CA curve arises from de-focusing and occurs with a negative nonlinear refractive index (*n*_2_ < 0).

Figure 2b shows a typical Z-scan setup [62]. To measure the ultrafast nonlinear response, a femtosecond pulsed laser is used to excite the samples. A half-wave plate combined with a linear polarizer can be employed to control the power of the incident light. The beam is focused onto the sample with a lens or an objective. During the measurements, samples are oriented perpendicular to the beam axis and translated along the Z axis with a linear motorized stage. For the measurements of small micrometer sized samples, a high-definition charge-coupled-device imaging system can be employed to align the light beam to the target area. Two PDs are employed to detect the transmitted light power for the signal and reference arms.

For the CA Z-scan method, the normalized transmittance can be written as [62,104]:(4)$$T\;\left(z,\;∆\Phi_{0}\right)≃1+\frac{4∆\Phi_{0}x}{\left(x^{2}+9\right)\left(x^{2}+1\right)}$$
where $x=z/z_{0}$, $z_{0}=k\omega_{0}^{2}/2$ with $\omega_{0}$ the beam waist radius and *k* the wave vector. $∆\Phi_{0}\;$represents the on-axis phase shift at the focus, is defined as [62,104]:(5)$$∆\Phi_{0}=kn_{2}I_{0}L_{eff}$$

In Equation (5), $L_{eff}=\left(1-e^{-\alpha L}\right)/\alpha$, with *L* denoting the sample length and $\alpha_{0}$ the linear absorption coefficient, *k* is the wave vector which is defined by k=2π/λ, and $I_{0}$ is the laser irradiance intensity with in the sample [104]. Based on the measured Z-scan curves, one can derive the Kerr coefficient *n*_2_ with the fitting equations.

Graphene is the first 2D material to have been discovered, and its optical nonlinearities have been widely studied using Z-scan measurements and other methods. Figure 3a shows the CA Z-scan signal of a graphene film with an excitation laser wavelength at 1550 nm [105]. A peak–valley configuration can be observed, indicating a negative Kerr nonlinearity. The measured Kerr coefficient *n*_2_ of graphene is as large as 10^−11^ m^2^/W which is about six orders of magnitude larger than bulk Si, demonstrating the strong potential of 2D materials for nonlinear optical devices. A laser peak intensity-dependent *n*_2_ has also been observed (Figure 3b), providing a potential method for modulating its nonlinear properties. Figure 3c,d show the CA curves of CH_3_NH_3_PbI_3_ perovskite [106] and Ti_3_C_2_T_x_ MXene films [57] measured at a wavelength of 1550 nm, where a positive and negative Kerr nonlinearity were observed, respectively. The different response of these two materials forms the basis of their applications in different functional devices. For example, a negative Kerr nonlinearity can be used to self-compress ultrashort pulses in the presence of positive group-velocity dispersion while the materials with positive nonlinearity are promising for achieving a net parametric modulational instability gain under abnormal dispersion conditions.

2D van der Waals (vdW) heterostructures offer many new features and possibilities beyond what a single material can provide, and there has been significant activity in this field [107,108,109]. Recently, the optical nonlinear response of 2D heterostructures has also been investigated via the Z-scan method. Figure 3e plots the CA curve of a MoS_2_/BP/MoS_2_ heterostructure at different laser intensities [107]. A negative Kerr nonlinearity at the telecommunications wavelength of 1550 nm can be observed. The strong Kerr nonlinearity of graphene/Bi_2_Te_3_ at the same wavelength was also demonstrated recently [110]. By fitting the experimental data, a large *n*_2_ of ∼2 × 10^−12^ m^2^/W was obtained, which is highly attractive for all-optical modulators and switches.

One of the unique features of GO is its tunable optical and electrical properties through laser reduction, which is particularly attractive for nonlinear optical applications. To investigate laser tunable optical nonlinearities, an in situ third-order Kerr nonlinearity measurement for GO films has been conducted with the Z-scan method [60]. Figure 4a–d show the CA signal of GO films at different laser intensities. At low intensity, GO exhibits a positive Kerr nonlinearity with a valley–peak CA configuration. With increasing the laser intensity, GO reduction occurs and the positive nonlinearity finally transitions into a negative nonlinearity at an intensity of 4.63 GW/cm^2^, at which point GO completely reduces to graphene. In addition to the ability to laser tune optical nonlinearities in GO, the measured Kerr coefficient *n*_2_ of GO is as large as 4.5 × 10^−14^ m^2^/W at 1550 nm, which is four orders of magnitude higher than single crystalline silicon. These properties render GO a promising candidate for nonlinear applications in the telecommunications band.

#### 3.2.2. THG Measurement

In addition to the Z-scan method, another technique that can be used to directly characterize the third-order optical nonlinearity of a material is THG measurement. As introduced in Section 3.1, THG is a fundamental third-order optical nonlinear process in which three photons at the same frequency (*ω*_1_) excite the nonlinear media to generate new signal (*ω* = 3*ω*_1_). Measuring the THG of a material provides a direct method to characterize its third-order optical nonlinearity. Figure 5 shows a typical setup for THG measurements [111] where a fundamental (ω, red) pulse is incident normally on the sample. The third harmonic (3ω, green) is detected in the reflected direction by a CCD camera, a spectrometer, or a photodiode connected to a lock-in amplifier.

To quantitatively analyze the THG effect, an equation for the THG intensity ($I_{3\omega })$, can be introduced [112]:(6)$$I_{3\omega }\left(t\right)=\frac{9\omega^{2}}{16\left|{\tilde{n}}_{3\omega }\right|{\left|{\tilde{n}}_{\omega }\right|}^{3}\epsilon_{0}^{2}c^{4}}I_{\omega }^{3}{\left|\chi^{\left(3\right)}\right|}^{2}\left(\frac{e^{-2\alpha t}-2\cos\left(\Delta kt\right)e^{-\alpha t}+1}{\alpha^{2}+\Delta k^{2}}\right)e^{-2\alpha t}$$
where ${\tilde{n}}_{\omega }\;$and ${\tilde{n}}_{3\omega }$ are the complex refractive indexes at the fundamental and harmonic wavelengths, respectively, α is the absorption coefficient at the THG wavelength, Δ*k* is the phase mismatch between the fundamental and harmonic waves, and $\chi^{\left(3\right)}\;$is the third-order susceptibility of the sample. By fitting the THG data with Equation (6), an effective third-order susceptibility χ^(3)^ value can be obtained.

Strong THG in graphene was demonstrated by Kumar et al. [111]. Figure 6a-i show the THG of monolayer graphene as a function of incident laser powers. The incident laser was 1720.4 nm. By fitting the experimental data, a large χ^(3)^ of ∼0.4 × 10^−16^ m^2^/V^2^ was obtained. In addition, a thickness-dependent THG signal can be observed (Figure 6a-ii, while χ^(3)^ remains constant with increasing graphene layer number. Recently, Jiang et al. [113] investigated the gate-tunable THG of graphene. Figure 6b-ii show the THG signal as a function of chemical potential generated at different wavelengths. When tuning the doping level of graphene, an enhanced THG and χ^(3)^ were observed.

THG in other 2D materials, such as TMDCs and BP, have also been investigated recently. Rosa et al. [114] characterized THG in mechanically exfoliated WSe_2_ flakes at an excitation wavelength of 1560 nm. By measuring the THG for different numbers of layers, a clear thickness-dependent behaviour was observed, as shown in Figure 7a-i,a-iii. The χ^3)^ of WSe_2_ was measured to be in the order of 10^−19^ m^2^/V^2^, which is comparable to other TMD [115] and BP [116]. Youngblood et al. [116] reported THG in BP by using an ultrafast near-IR laser obtaining a χ^(3)^ of ∼1.4 × 10^−19^ m^2^/V^2^. In addition, an anisotropic THG was demonstrated, as shown in Figure 7b-iii. Nonlinear optical properties of few-layer GaTe were also studied by characterizing the THG at a pump wavelength of 1560 nm [117]. The THG intensity was found to be sensitive to the number of GaTe layers (Figure 7c-iii). They obtained a large χ^(3)^ of ∼2 × 10^−16^ m^2^/V^2^

#### 3.2.3. Hybrid Device Characterization

Z-scan and THG measurements are usually employed to characterize the material property directly. While on the one hand, the properties of a material form the basis for applications to electronic and optical devices, the reverse is true—device performance can also provide key information about the material properties. A typical example is field effect transistors (FETs) which have been one of the main techniques to evaluate the electrical properties of 2D materials. Optical structures and waveguides can also be exploited to characterize the material optical properties. By integrating 2D materials with photonic cavities and optical waveguides, the third-order optical nonlinearity of atomically thin 2D material has been characterized by measuring the nonlinear optical responses of the hybrid devices, such as FWM [78], SPM [99], and supercontinuum generation [118]. This method also enables the investigation of the layer-dependence of the nonlinear properties, which is challenging for conventional Z-scan methods due to the weak response of ultrathin 2D films.

For the hybrid device characterization, the data analysis is performed in the following steps. First, by fitting the measured FWM or SPM spectra of corresponding hybrid devices, one can obtain the nonlinear parameters (*γ*) for the bare and hybrid waveguides. Then, based on the fit *γ* of the hybrid waveguides, the Kerr coefficient (*n*_2_) of the coated 2D films can be extracted using [119,120,121]:(7)$$\gamma =\frac{2\pi }{\lambda \;}\frac{\iint_{D}^{}n_{0}{}^{2}\left(x,\;y\right)n_{2}\left(x,\;y\right)S_{z}{}^{2}dxdy}{{\left[\iint_{D}^{}n_{0}\left(x,\;y\right)S_{z}dxdy\right]}^{2}}$$
where *λ* is the central wavelength, *D* is the integral of the optical fields over the material regions, *S_z_* is the time-averaged Poynting vector calculated using mode solving software, *n*_0_ (*x*, *y*) and *n*_2_ (*x*, *y*) are the refractive index profiles calculated over the waveguide cross section and the Kerr coefficient of the different material regions, respectively.

FWM is a fundamental third-order nonlinear optical process that has been widely used for all optical signal generation and processing, including wavelength conversion [98,122], optical frequency comb generation [123,124], optical sampling [125,126], quantum entanglement [29,30], and many other processes. The conversion efficiency (CE) of FWM is mainly determined by the third-order Kerr nonlinearity of the material that makes of the device. Therefore, it is useful to obtain the Kerr coefficient of a material by measuring its FWM CE.

Gu et al. [118] fabricated a silicon nanocavity covered with graphene (Figure 8a-i) and measured the FWM CE with different pump and signal detuning wavelengths around 1550 nm, as shown in Figure 8a-ii,a-iii. From the CE data, a *n*_2_ of ∼4.8 × 10^−17^ m^2^/W was obtained for a graphene integrated with a silicon cavity. The layer-dependence of the Kerr nonlinearity of GO films has been investigated by measuring the FWM performance of GO hybrid devices based on doped-silica and SiN optical waveguides and microring resonators (MRRs) [78,127,128,129]. Figure 8b-i show a fabricated doped-silica MRR covered with patterned GO films [78]. By fitting the CE to theory for a device with different GO thicknesses, the layer thickness dependence of *n*_2_ of GO at 1550 nm was characterized, as shown in Figure 8b-iii. Recently, electrically tuneable optical nonlinearities of graphene at 1550 nm were also demonstrated by measuring FWM in graphene-SiN waveguides at different gate voltages, as shown in Figure 8c [130].

SPM is another third-order nonlinear optical process that can be used to characterize the optical nonlinearity of 2D materials. Feng et al. [131] studied the Kerr nonlinearities of graphene/Si hybrid waveguides with enhanced SPM (Figure 9a). The *n*_2_ of the Graphene on Si hybrid waveguides was measured to be ∼2 × 10^−17^ m^2^/W, which is three times larger than that of the Si waveguide. Even though the intrinsic *n*_2_ of graphene is orders of magnitude larger than bulk silicon, the monolayer thickness of the graphene film results in a very low optical mode overlap, which yields only a factor of three improvement in the effective nonlinearity of the waveguide. For GO, on the other hand, comparatively larger film thicknesses are achievable which result in an overall much higher waveguide nonlinearity. Optical nonlinearities of GO films have also been investigated by SPM experiments. Zhang et al. [99] demonstrated the enhanced optical nonlinearity of silicon nanowires integrated with 2D GO Films (Figure 9b-i). Figure 9b-ii show the experimental SPM spectra of the devices with different numbers of GO layers, where increased spectral broadening can be observed in GO coated silicon nanowires. The layer-dependent Kerr *n*_2_ coefficient of GO was also characterized by fitting the spectra to theory, as shown in Figure 9b-iii. In addition to graphene and GO, the optical Kerr nonlinearity of MoS_2_ monolayer films was also characterized by analysing the SPM of MoS_2_-silicon waveguides [132]. The experiments demonstrated a large Kerr coefficient *n*_2_ of ∼1.1 × 10^−16^ m^2^/W for a monolayer of MoS_2_ in the telecommunications band.

### 3.3. Comparison of Measured Results

By using different characterization techniques discussed above, Kerr coefficient *n*_2_ or THG $\chi^{\left(3\right)}$ value of graphene, GO, TMDCs, BP, and different heterostructures at telecommunication wavelengths have been obtained. In Table 1, we compare these parameters characterizing the third-order optical nonlinearity. It can be seen that monolayer graphene exhibits the largest *n*_2_ value (up to 10^−11^ m^2^/W). The *n*_2_ value of GO films is on the magnitude of 10^14^ m^2^/W, which is relatively smaller than graphene, but still more than three orders of magnitudes larger than that of bulk silicon. For MoS_2_, MXene film, and 2D heterostructures, the measured *n*_2_ varies from 10^−16^ to 10^−22^ m^2^/W. In terms of $\chi^{\left(3\right)}$ susceptivity obtained by using THG measurements, the value ranges from 10^−19^ to 10^−15^ m^2^/V^2^ for graphene, TMDCs, and BP.

One should note that the measured *n*_2_ values can often vary even for the same material with the use of different measurement techniques, mainly due to the difference in sample preparation and different laser sources used in the measurements. Usually, 2D films fabricated by chemical vapor deposition possess higher numbers of defects compared to mechanically exfoliated single crystal monolayers [37,38], and this can often be reflected in variations in the measured values of *n*_2_. As for the influence of irradiation lasers, the pulse duration is a key factor. In Z-scan and THG measured, a femtosecond laser is widely employed while a picosecond laser or continuous wavelength (CW) laser are used in the characterization of hybrid devices. A longer laser duration, particularly for the CW laser, may induce thermal optical nonlinearity of the materials, resulting in the deviation of the measured *n*_2_ values.

## 4. Outlook and Prospects

The past decade has witnessed tremendous progress of 2D materials in both fundamental property study and related device applications. As for optical applications, the excellent third-order optical nonlinearity of graphene, GO, TMDCs, and many other novel 2D materials have been investigated using various techniques, and these form the basis of their applications in high-performance nonlinear photonic devices for next-generation optical communications systems.

Despite these remarkable achievements, challenges still exist for engineering the nonlinear optical properties of 2D materials. First, accurate and efficient characterization of the linear and nonlinear optical properties remains challenging. Although the Z-scan method has been highly successful, the very weak Z-scan signals of ultra-thin films limit its applications in mono- or few layer 2D materials, especially for the layer-dependent measurements. In contrast, integrating 2D films with optical waveguides provides a powerful method to obtain accurate nonlinear parameters of atomic-thin 2D materials by analyzing the nonlinear optical performance of the hybrid device. However, the complicated device fabrication process and resulting relatively low efficiency make this method unsuitable for rapid material characterization which is required for future industrial applications. Second, 2D materials are a large family which include thousands of different materials. For applications of the third-order optical nonlinearity in the telecommunications band, only a very small fraction of them have been investigated. Many newer materials, such as perovskites, MOFs, and graphdiyne, still need more research, which hinders the full exploitation of 2D materials in the fabrication of next-generation nonlinear optical devices. Finally, tuning or engineering the properties of materials is important for both optimizing the device performance and enabling new functionalities, as well as the fundamental study of 2D materials. Nevertheless, current advances in the study of third-order optical nonlinearities of 2D materials focus mainly on their fundamental properties. The relative lack of effective methods of tuning the material properties poses another obstacle for 2D materials to move forward to practical device fabrication. While challenges remain and more work is needed, there is no doubt that 2D materials will underpin key breakthroughs and greatly accelerate the developments of next-generation nonlinear optical devices for many applications, particularly high bandwidth optical communications systems.

## 5. Conclusions

In conclusion, we review recent progress in the study of the third-order optical nonlinearities of 2D materials in the telecommunications wavelength band. We introduce the representative 2D materials, together with their basic material properties followed by a discussion of the main methods for characterizing the third-order optical nonlinearity, reviewing recent achievements in the field. These advances highlight the significant potential of 2D materials in enabling high-performance nonlinear optical devices for all-optical processing functions in optical communications systems.

## Figures and Tables

**Figure 1 micromachines-14-00307-f001:**
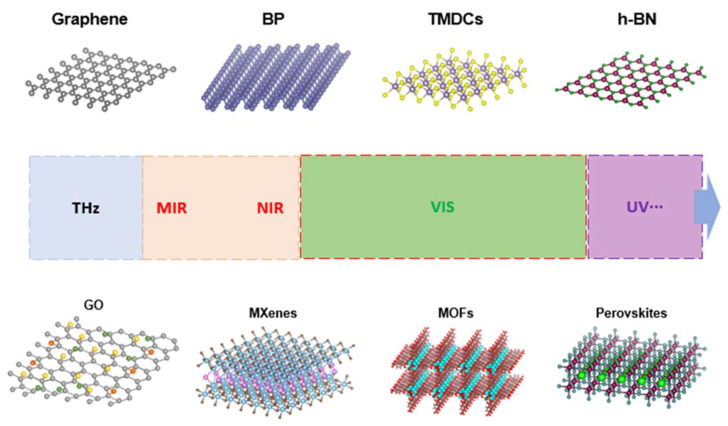
Illustration of typical 2D-layered materials.

**Figure 2 micromachines-14-00307-f002:**
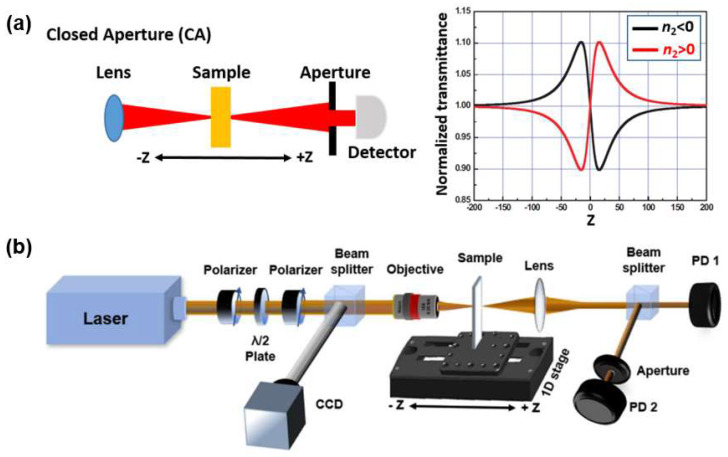
(**a**) Schemes showing the principle of closed-aperture (CA) Z-scan. (**b**) A typical Z-scan setup: PD: power detector, CCD: charge-coupled-device [62].

**Figure 3 micromachines-14-00307-f003:**
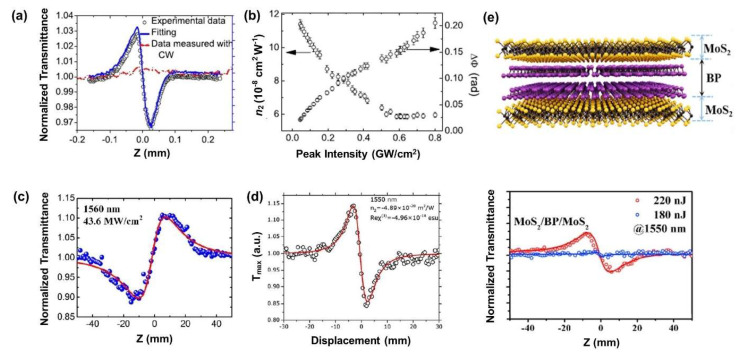
(**a**) CA Z-scan result of graphene under an excitation laser wavelength at 1550 nm. (**b**) Measured *n*_2_ of graphene as a function of laser intensity [105]. (**c**) CA Z-scan result of CH_3_NH_3_PbI_3_ under an excitation laser wavelength at 1560 nm [106]. (**d**) CA Z-scan result of MXene films under an excitation laser wavelength at 1550 nm [57]. (**e**) CA Z-scan result of MoS_2_/BP/MoS_2_ heterostructure at different laser intensities. The excitation laser wavelength is 1550 nm [107].

**Figure 4 micromachines-14-00307-f004:**
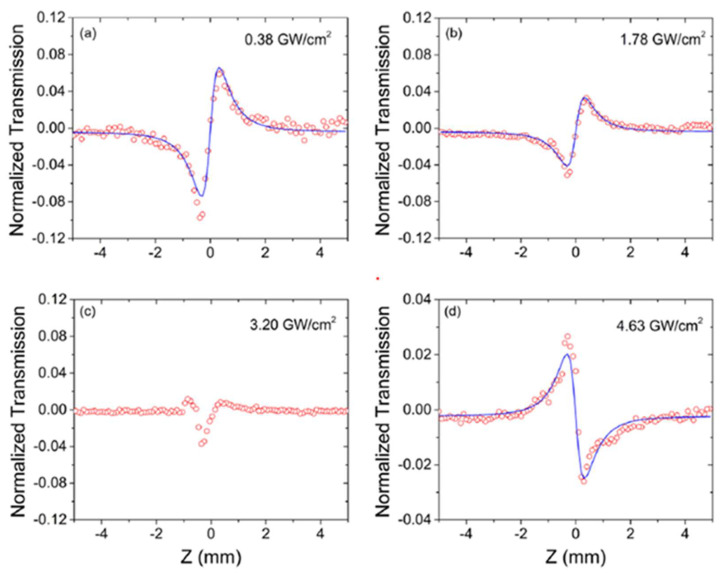
The CA Z-scan results of GO films under different irradiances: (**a**) 0.38 GW/cm^2^; (**b**) 1.78 GW/cm^2^; (**c**) 3.20 GW/cm^2^; (**d**) 4.68 GW/cm^2^ [60].

**Figure 5 micromachines-14-00307-f005:**
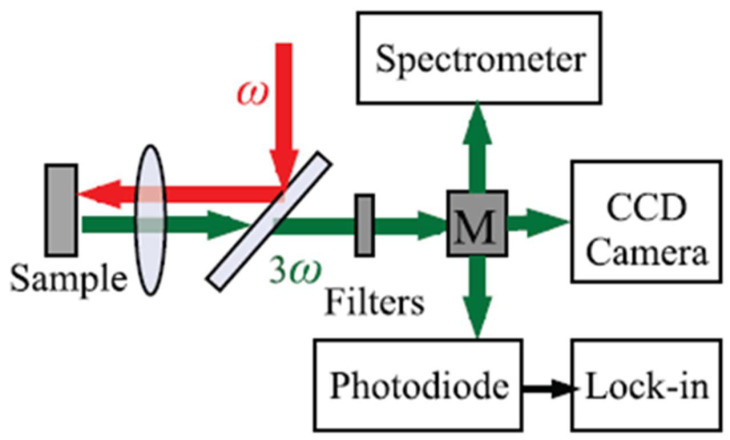
The scheme of a typical THG setup: CCD, charge-coupled-device [111].

**Figure 6 micromachines-14-00307-f006:**
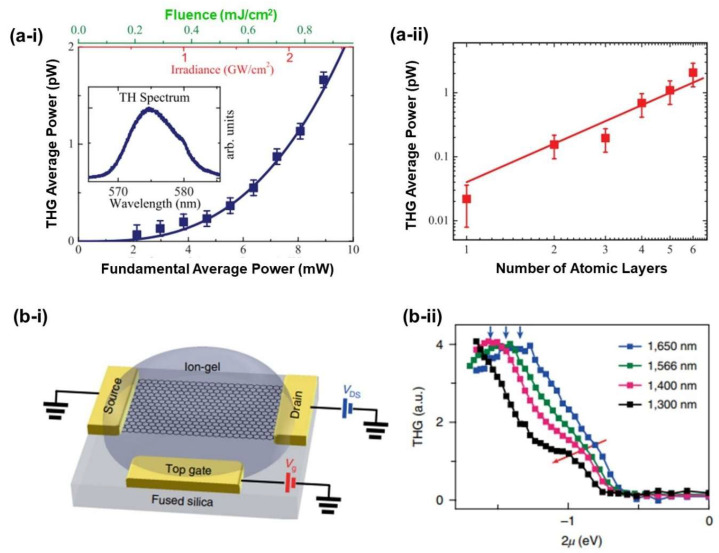
(**a**) THG in graphene: [111] (**a-i**) The average power of the THG signals as a function of the average power of the incident laser. Inset is the THG spectrum. (**a-ii**) The average power of the THG signals as a function of the number of atomic layers for an average fundamental power of 1 mW. (**b**) Gate-tunable THG in graphene: [113] (**b-i**) Schematic of an ion-gel-gated graphene monolayer on a fused silica substrate covered by ion-gel and voltage biased by the top gate. (**b-ii**) THG signal as a function of 2 *μ* generated by different input wavelengths: 1300 nm, 1400 nm, 1566 nm and 1650 nm.

**Figure 7 micromachines-14-00307-f007:**
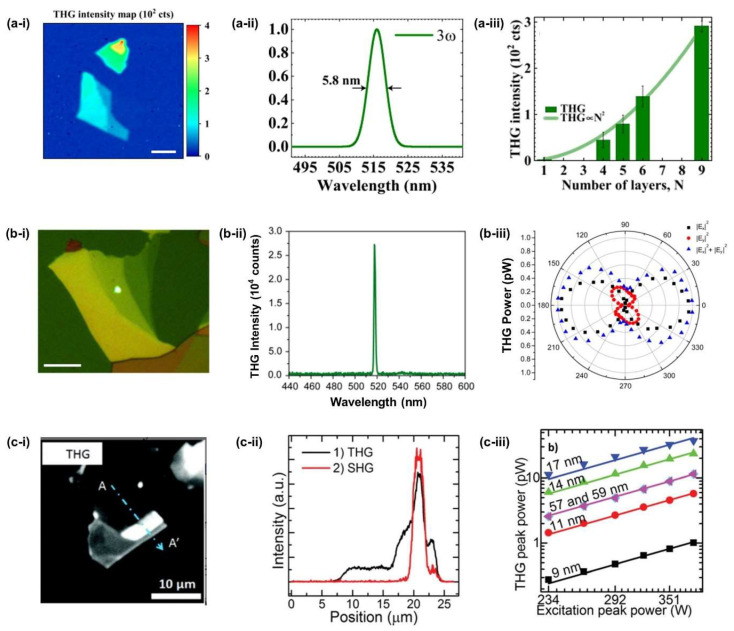
(**a**) THG in WSe_2_: [114] (**a-i**) Spatial THG intensity mapping across the WSe_2_ sample. (**a-ii**) THG spectrum of WSe_2_. (**a-iii**) THG intensities as a function of sample layers. (**b**) THG in BP: [116] (**b-i**) THG emission (bright spot) from the BP flake. (**b-ii**) Measured spectrum of THG emission with a peak wavelength at 519 nm. (**b-iii**) Anisotropic THG in BP. (**c**) THG in GaTe: [117] (**c-i**) THG images of the few-layer GaTe flake. (**c-ii**) Measured spectra of THG emission. (**c-iii**) THG signals of samples with different thicknesses.

**Figure 8 micromachines-14-00307-f008:**
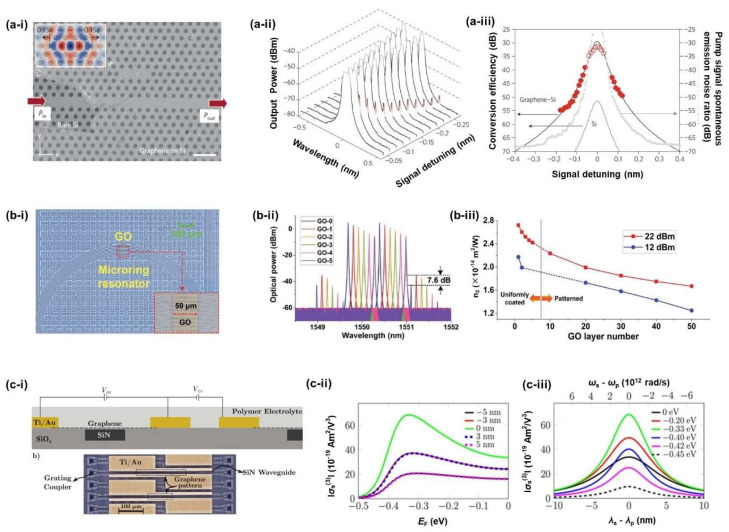
(**a**) FWM in graphene-clad silicon nanocavities: [118] (**a-i**) Scanning electron micrograph (SEM) of the photonic crystal cavity partially covered by graphene monolayer. (**a-ii**) Measured transmission spectrum of the cavity device with pump laser fixed on cavity resonance, and signal laser detuning scanned from 20.04 to 20.27 nm. (**a-iii**) Measured and simulated conversion efficiencies of the cavity. Solid and dashed black lines are modelled conversion efficiencies for the graphene–silicon and monolithic silicon cavities, respectively. (**b**) Layer-dependent optical nonlinearities in GO-coated integrated MRR: [78] (**b-i**) Microscopic image of an integrated MRR patterned with 50 layers of GO. Inset shows zoom-in view of the patterned GO film. (**b-ii**) Optical spectra of FWM at a pump power of 22 dBm for the MRRs with 1−5 layers coated GO. (**b-iii**) *n*_2_ of GO versus layer number at fixed pump powers of 12 and 22 dBm. (**c**) Electrically tunable optical nonlinearities in graphene-covered SiN Waveguides: [130] (**c-i**) Sketch of the gating scheme (up) and optical microscope image (down) of the device. Calculated values of third-order conductivity as a function of Fermi energy for different wavelength detunings (**c-ii**) and as a function of detuning for a range of Fermi energies (**c-iii**).

**Figure 9 micromachines-14-00307-f009:**
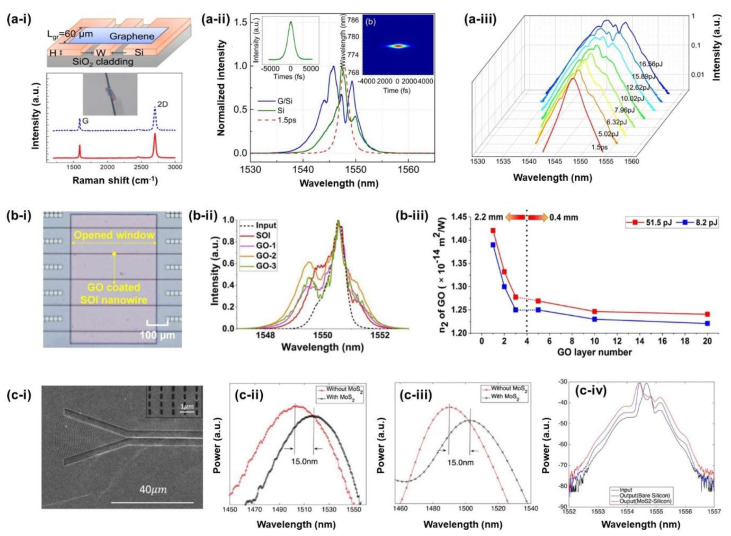
(**a**) SPM experiments in graphene–silicon hybrid waveguides: [131] (**a-i**) Schematic diagram and Raman spectra of the device. (**a-ii**) Measured transmission spectra of comparison between the Si (green solid curve) and G/Si hybrid (blue solid curve) waveguides under the same input energy with 1.5 ps input pulse (spectrum denoted by the red dashed curve). (**a-iii**) Output SPM spectra of the hybrid waveguide under various coupled energies. (**b**) SPM experiments in GO-silicon waveguide: [99] (**b-i**) Microscopic image of a GO-coated silicon nanowire. (**b-ii**) Optical spectra of SPM at a coupled pulse energy of ∼51.5 pJ with 1−3 layers coated GO. (**b-iii**) *n*_2_ of GO vs. layer number at fixed coupled pulse energies of 8.2 and 51.5 pJ. (**c**) SPM experiments in MoS_2_-coated silicon waveguides: [132] (**c-i**) scanning electron microscope image of the device, with MoS_2_ covering both the grating couplers and waveguide regions. (**c-ii**) Measured transmission spectra of the devices with and without MoS_2_. (**c-iii**) Simulation result for the redshift of the grating to estimate the refractive index of MoS_2_. (**c-iv**) SPM spectra of MoS_2_–silicon waveguide and bare silicon waveguide.

**Table 1 micromachines-14-00307-t001:** Comparison of third-order optical nonlinear parameters of different 2D materials. FWM: four-wave mixing; SPM: self-phase modulation; WG: waveguide; MRR: microring resonator; THG: third-harmonic generation.

Material	Wavelength ^(a)^	Thickness	Nonlinear Parameter	Method	Ref.
Graphene	1550 nm	∼1 layer	*n*_2_ = ∼10^−11^ m^2^/W	Z-scan	[105]
Graphene	1550 nm	∼5–7 layers	*n*_2_ = ∼−8 × 10^−14^ m^2^/W	Z-scan	[58]
GO	1560 nm	∼1 um	*n*_2_ = ∼4.5 × 10^−14^ m^2^/W	Z-scan	[60]
GeP	1550 nm	∼15–40 nm	*n*_2_ = ∼3.3 × 10^−19^ m^2^/W	Z-scan	[133]
CH_3_NH_3_PbI_3_	1560 nm	∼180 nm	*n*_2_ = ∼1.6 × 10^−12^ m^2^/W	Z-scan	[106]
MXene	1550 nm	∼220 um	*n*_2_ = ∼−4.89 × 10^−20^ m^2^/W	Z-scan	[57]
BiOBr	1550 nm	∼140 nm	*n*_2_ = ∼3.82 × 10^−14^ m^2^/W	Z-scan	[61]
MOF	1550 nm	∼4.2 nm	*n*_2_ = ∼−8.9 × 10^−20^ m^2^/W	Z-scan	[134]
MoS_2_/BP/MoS_2_	1550 nm	∼17–20 nm	*n*_2_ = ∼3.04 × 10^−22^ m^2^/W	Z-scan	[107]
Graphene/Bi_2_Te_3_	1550 nm	∼8.5 nm	*n*_2_ = ∼2 × 10^−12^ m^2^/W	Z-scan	[110]
Graphene	1550 nm	∼1 layer	*n*_2_ = ∼10^−13^ m^2^/W	SPM in WG	[135]
GO	1550 nm	∼4 nm	*n*_2_ = ∼1.5 × 10^−14^ m^2^/W	FWM in WG	[127]
GO	1550 nm	∼2–100 nm	*n*_2_ = ∼(1.2-2.7) × 10^−14^ m^2^/W	FWM in MRR	[78]
GO	1550 nm	∼2–20 nm	*n*_2_ = ∼(1.3-1.4) × 10^−14^ m^2^/W	FWM in WG	[128]
GO	1550 nm	∼2–40 nm	*n*_2_ = ∼(1.2-1.4) × 10^−14^ m^2^/W	SPM in WG	[99]
MoS_2_	1550 nm	∼1 layer	*n*_2_ = ∼1.1 × 10^−16^ m^2^/W	SPM in WG	[132]
Graphene	1560 nm	∼1 layer	χ^(3)^ = ∼4 × 10^−15^ m^2^/V^2^	THG	[136]
Graphene	1560 nm	∼1 layer	χ^(3)^ = ∼1.5 × 10^−19^ m^2^/V^2^	THG	[137]
MoS_2_	1560 nm	∼1 layer	χ^(3)^ = ∼2.4 × 10^−19^ m^2^/V^2^	THG	[137]
MoSe_2_	1560 nm	∼1 layer	χ^(3)^ = ∼2.2 × 10^−19^ m^2^/V^2^	THG	[138]
WS_2_	1560 nm	∼1 layer	χ^(3)^ = ∼2.4 × 10^−19^ m^2^/V^2^	THG	[138]
WSe_2_	1560 nm	∼1 layer	χ^(3)^ = ∼1.2 × 10^−19^ m^2^/V^2^	THG	[114]
SnSe_2_	1560 nm	multilayer	χ^(3)^ = ∼4.1 × 10^−19^ m^2^/V^2^	THG	[139]
ReS_2_	1515 nm	∼1 layer	χ^(3)^ = ∼5.3 × 10^−18^ m^2^/V^2^	THG	[140]
BP	1560 nm	multilayer	Χ(^3)^ = ∼1.6 × 10^−19^ m^2^/V^2^	THG	[141]

^(a)^ Here, is the excitation laser wavelength.

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
