# Peer review of "Third-Order Optical Nonlinearities of 2D Materials at Telecommunications Wavelengths"

_micromachines, 2023, doi:10.3390/mi14020307_

Round 1

Reviewer 1 Report

In this paper, : the authors  review recent advances in research on third-order optical nonlinearities of 2D materials, focusing on all-optical processing applications in the optical telecommunications band near 1550 nm. The topic  is interesting . However, some issues should be addressed.

·       In the introduction, the authors should clearly highlight the novelty of their work in comparison to previous works.

·       The conclusion section should be extended and improved to help a reader not familiar with the topic.

·       In the introduction, the authors should focus on the fundamental properties of 2D materials

see for example

1 Variable angle spectroscopic ellipsometry characterization of spin-coated MoS2 films,

Vacuum, Volume 189,2021,110232,

2 A Review of the Synthesis, Properties, and Applications of 2D Materials. Part. Part. Syst. Charact. 2022, 39, 2200031. https://doi.org/10.1002/ppsc.202200031

3 J Munuera et al 2022 2D Mater. 9 012002

Author Response

Detailed response to comments from reviewers

Title: Third-order optical nonlinearities of 2D materials at telecommunications wavelengths

We thank the reviewers for their comments. We address their comments in detail here and have marked the changes in red in the revised manuscript accordingly.

Blue Italic: Original comments from reviewer. Black: Our response including action taken.

Comments from Reviewer 1:

In this paper, the authors review recent advances in research on third-order optical nonlinearities of 2D materials, focusing on all-optical processing applications in the optical telecommunications band near 1550 nm. The topic is interesting. However, some issues should be addressed.:

1 In the introduction, the authors should clearly highlight the novelty of their work in comparison to previous works.

We thank the reviewer for the positive comments. Optical materials with high third-order optical nonlinearity form the vital basis for implementing high-performance nonlinear optical devices in optical communication systems. Among them, 2D materials have attracted intense interests. Though previous reviews have generally summarized the progress of nonlinear optics based on 2D materials, they mainly focused on the visible wavelengths. A systematic review of their properties at telecommunication wavelengths and related characterization techniques has not been conducted. Our review is helpful for readers to have a glance of the progress in this field and will provide guidance for the future material optimization and device applications of 2D materials in optical telecommunication systems.   

To address this issue, we have added related comments in the revised manuscript as follows:

Page 2, Line 62

“In this paper, we review recent progress in the study of the third-order optical nonlinearities of 2D materials specifically in the telecommunications wavelength band near 1550 nm, in contrast with other reviews [44, 69, 70] that focus on the visible wavelength range.”

Page 2, Line 72

“Our review is aimed to help readers have a view of the progress in this field and to provide guidance for optimizing the material properties and device applications of 2D materials in future optical telecommunications systems.”

  1. The conclusion section should be extended and improved to help a reader not familiar with the topic.

As suggested, we have added related comments in the revised manuscript as follows:

Page 14, Line 449

“The past decade has witnessed tremendous progress of 2D materials in both the fundamental studies of material properties as well as related device applications. As for the optical applications, the excellent third-order optical nonlinearities of graphene, GO, TMDCs, and many other novel 2D materials have been investigated by using various techniques, which forms the basis of their applications in high-performance nonlinear photonic devices for next-generation optical communications systems.”

  1. In the introduction, the authors should focus on the fundamental properties of 2D materials

We thank the reviewer’s suggestion. We have added more comments about the fundamental properties of 2D materials as follows:

Page 1, Line 43

“Compared with bulk materials, the 2D materials possess surfaces that are free of dangling bonds due to their weak out-of-plane van der Waals interactions [32, 37].”

Page 2, Line 50

“The unique photon-excited exciton and valley-selective properties of monolayer TMDCs and their heterostructures are promising for the development of future spintronic and quantum computing devices [37, 38].”

Comments from Reviewer 2:

  1. This review is about third-order non-resonant nonlinearities. This could be reflected in the title to clearly identify the publication "Non-resonant Third-order optical nonlinearities of 2D materials at telecom wavelengths".

We thank the reviewer for the suggestion. Our review focuses on the third-order nonlinear optical properties of 2D materials at telecommunication wavelengths and related characterization methods. Although the third-order optical nonlinearities mentioned in our review are measured with non-resonant structures, in-depth discussion on the physics of the resonant or non-resonant optical nonlinearities is not included. Therefore, we think the current title is more suitable and want to keep it if possible.

  1. Can the author comment on the impact of the laser pulse used in the n² measurements?

In the revised manuscript, we have added related comment to address this issue as follows:

Page 13, Line 448

“As for the influence of the optical pulses of the lasers used for irradiation, the pulse duration is a key factor. In Z-scan and THG measurements, a femtosecond laser is widely employed while picosecond laser or even continuous wavelength (CW) lasers are used in the characterization of hybrid devices. Longer pulsed lasers, and particularly for CW lasers, can often induce thermal optical non-linearities of the material, resulting in the variations in the effective measured values of n2.”

  1. Can the author introduce a paragraph to comment on the variability of the measured n² depending on the method used?

In the revised manuscript, we have added related comment to address this issue as follows:

Page 13, Line 448

“One should note that the measured n2 values can often vary even for the same material with the use of different measurement techniques, mainly due to the difference in sample preparation and different laser sources used in the measurements. Usually, 2D films fabricated by chemical vapor deposition possess higher numbers of defects compared to mechanically exfoliated single crystal monolayers [37, 38], and this can often be reflected in variations in the measured values of n2.”

Reviewer 2 Report

1) This review is about third-order non-resonant nonlinearities. This could be reflected in the title to clearly identify the publication "Non-resonant Third-order optical nonlinearities of 2D materials at telecom wavelengths".

Can the author comment on the impact of the laser pulse used in the n² measurements?

Can the author introduce a paragraph to comment on the variability of the measured n² depending on the method used?

Author Response

(The authors gave the same response as above.)
